# Subjective Well-Being and Parenthood in Chile

**DOI:** 10.3390/ijerph18147408

**Published:** 2021-07-11

**Authors:** Consuelo Novoa, Claudio Bustos, Vasily Bühring, Karen Oliva, Darío Páez, Pablo Vergara-Barra, Félix Cova

**Affiliations:** 1Department of Psychology, Universidad de Concepción, Concepcion 4070386, Chile; consuelonovoa@udec.cl (C.N.); clbustos@udec.cl (C.B.); vasilybuhring@udec.cl (V.B.); karenoliva@udec.cl (K.O.); 2Departament of Social Psychology and Methodology, Universidad del País Vasco, 20018 Leioa, Spain; dario.paez@ehu.es; 3Faculty of Education and Social Sciences, Universidad Andrés Bello, Santiago de Chile 7591538, Chile; 4Department of Psychiatry and Mental Health, Universidad de Concepción, Concepcion 4030000, Chile; pablovergara@udec.cl

**Keywords:** parenting, motherhood, fatherhood, children, happiness

## Abstract

Being a parent plays an important role in people’s life trajectory and identity. Though the general cultural perception is that having children is a source of subjective well-being, there is evidence that, at least in some societies, the subjective well-being of those who are parents is worse, in some aspects, than that of those who are not. This gap has been the object of interest and controversy. The aim of this study was to compare Chilean adults with and without children in a broad set of well-being indicators, controlling for other sociodemographic variables. A public national probabilistic database was used. The results show that, in terms of positive and negative affect, those who are not parents achieve greater well-being than those who have children. Other results also pointed in that direction. The implications of the social context and gender, which are aspects that pose a burden for the exercise of parenthood in Chile, are discussed.

## 1. Introduction

The intense transformations of social and family life in contemporary societies [1,2] and the conditions and ways in which parenthood is exercised have driven research on the relationship between subjective well-being and being a parent [3,4,5]. These transformations are particularly visible in the marked decrease in birth rates in most countries. Today, people have fewer children, at later ages, and it is more common for some people to decide not to have any [6,7]. On the other hand, social expectations and demands regarding parenthood have expanded, a change that has not necessarily been accompanied by conditions that favor its exercise [8].

Being a parent has an important meaning in people’s life trajectory and identity. Traditionally, parenthood has been perceived as a rewarding and enriching experience. In fact, most people with children describe their status as parents as a relevant and valuable facet of their lives, and a decisive factor in their personal self-realization [9,10]. However, the current social context may conspire, on some levels, against the subjective well-being of those who are parents, an issue that is not always visible. In fact, parenthood is often idealized in social discourses, while the difficulties associated with the experience are minimized [11,12,13]. In fact, several investigations have indicated the existence of a personal well-being gap between those who are parents with respect to those who are not, which has been the subject of interest and controversy [14,15,16]. In various dimensions (stress, depression, anxiety, life satisfaction, and the daily frequency of positive and negative emotions), a gap in subjective well-being that favors those who are not parents has been observed [17,18,19]. The 2016 World Happiness Report devoted a chapter to this topic and confirmed the existence of this subjective well-being gap in those who are parents in most countries, although its effect was small [20]. This report ranked Chile in ninth place within a group of 66 countries with the largest gap. It has been observed that this well-being gap is greater in countries where social conditions in support of the exercise of parenthood are poorer and that it disappears in countries that have social and economic policies that favor the compatibility of family and work life and provide support for the care of children [18].

The parental role implies significant demands that, under certain circumstances, can negatively affect the well-being of parents. The concept of parental stress [21] attempts to capture this dimension of parenthood, and four main sources of parental stress are recognized: role overload, interpersonal conflict, role captivity, and inter-role conflict [22]. The ways of life and cultural dimensions of the dominant hypermodernity in some present-day societies have an effect on increasing their demanding nature, which has even led, in some social contexts, to parental burnout [23]. It is interesting that a greater presence of parental burnout can be observed in individualistic cultures [24].

Most of the research on the well-being of mothers and fathers has been carried out in high-income countries. One of the few studies conducted in Latin America (with samples from Brazil, Mexico, and Chile) showed a positive relationship, though weak, between parenthood and life satisfaction [25]. The study of the relationship between well-being and parenthood in Latin America is of particular interest given that, despite the economic, social, and political difficulties existing in the region, the levels of subjective well-being in the continent tend to be high and comparatively higher than those in other regions of the world with a similar set of socioeconomic indicators, an issue that has led some researchers to talk about the “Latin American paradox” [26]. The value given to interpersonal and family relationships is a factor that accounts for this apparent paradox.

In Chile, qualitative research has shown that Chilean parents possess strong feelings of regret and a sense that their parenthood is a burden, although they strongly value the meaning and worth of being a parent [27,28,29]. A qualitative study on children’s opinions points in the same direction: in the eyes of their children, parents appear stressed by the responsibilities of family, domestic, and work life and as “prisoners” of such demands. Children feel “indebted” to their parents, who seem exhausted to them, and cannot reward them in return for their efforts [8]. Chile is among the countries where the work overload for wage-earning households is the most intense, with most people declaring that they are unable to reach a balance between family and work, with wide gender differences in this respect [30,31].

A fundamental change in Chile in recent years has been the greater incorporation of women into the labor market, which has created new demands relative to childcare, challenging both family arrangements and the roles traditionally undertaken by men and women [32]. Women’s participation rate in the labor market has thus far reached 48.9% [33]. In general, Chilean mothers make significant efforts to adequately perform their remunerated activities and take care of their children, which are coupled with intense feelings of ambivalence and guilt [27,28]. This increasing incorporation of women into the labor market has not been followed by a significant redistribution of roles between mothers and fathers, which is a relevant factor that also impacts this issue [34,35,36]. The main caregiving tasks are still undertaken by female figures such as mothers and grandmothers [37]. Likewise, fathers continue to perform the role of providers as their main mandate, although some changes can be observed with respect to the past, such as a greater daily involvement and bond with their children [38,39]. It is common for parents to experience strong feelings of insecurity about their abilities to satisfactorily fulfill their role, as well as significant contradictions between this mandate and the experience of their own individuality [13,27].

In agreement with trends observed in various countries [40], the most negative experiences related to subjective well-being in mothers and fathers are observed in those who undertake the role of both caregiver and main provider without substantial support from the other parent. These single-parent families are mostly with single mothers and are more frequently affected by unfavorable socioeconomic conditions [19,41,42].

This set of factors makes the study of the well-being of Chilean parents of particular interest. The purpose of this study was to compare Chilean adults with and without children with a broad set of subjective well-being indicators, using the results of a survey by the United Nations Development Program (PNUD in Spanish) as the source [29].

## 2. Materials and Methods

A secondary analysis was carried out based on the public database of the study on subjective well-being carried out by PNUD published in 2012. Authorization for the use of the data was granted by PNUD through a letter of support.

### 2.1. Participants

This study used a stratified multi-stage sample design. For stratification, the regions the country is divided into and the urban–rural conditions were taken into account. Samples were collected in three stages: primary, city blocks; secondary, household; and tertiary, people 18 years of age and older. The total sample size was 2532 people. The age range was 18 to 99 years with a mean and median of 46 years. A total of 20.6% of the participants were between 18 and 29 years old (520), 37.3% (943) between 30 and 49, 29.5% (774) between 50 and 69, and 11.6% (295) over 70 years old; 37.32% (n = 945) were men and 62.68% (n = 1587) women. The socioeconomic distribution (using the categories of the Asociación de Institutos de Estudios de Mercado y Opinión de Chile—Chilean Association of Market and Opinion Studies Institutes) was as follows: ABC1 (high status), 7.46% (n = 189); C2 group (medium status), 11.89% (n = 301); group C3 (medium-low status), 32.9 (n = 833); group D (low status), 26.97% (n = 683); group E (poverty), 20.77 (n = 526). Additionally, 76.7% (n = 484) of participants had a partner, while 23.3% (n = 147) did not.

Of the total number of participants, 80.8% (n = 2044) reported having children and 19.2% (n = 486) did not. The average number of children of those who were parents was 2.73 (SD = 1.65) and the median was 2. The proportion of participants who were parents differed by socioeconomic status (ABC1 = 68%; C2 = 77%; C3 = 79%; D = 83%; E = 86%).

### 2.2. Variables and Instruments

For this study, information on parenthood (having children or not) and sociodemographic data (age, sex, socioeconomic status, having a partner) from the PNUD database were used. In this study, various questions and scales were used as indicators of well-being. Some of these indicators are broadly used in international research (life satisfaction, Cantril’s Ladder of Life PANAS) and have been recommended for use in Chile [43]. The other measurements correspond to items that were selected or developed to investigate specific topics considered in the PNUD study.

*Life satisfaction.* This indicator was measured through the question: “all things considered, how satisfied are you with your life right now?” (1 = completely dissatisfied; 10 = completely satisfied). The Better Life initiative [44] recommends the use of this measurement.

*The ladder of the best possible life.* This indicator was measured by the question: “Let us suppose that the highest step of the ladder represents the best possible life for you and the lowest one the worst possible life for you. On what step of the ladder do you feel you stand at present, where zero is the lowest and 10 is the highest?”. This question was based on Cantril’s Ladder of Life [45], a single item widely used to measure life satisfaction.

*Suffering.* This indicator was measured by the question: “How much suffering would you say there has been in your life?” (1 = no suffering; 10 = maximum suffering).

*Depressive symptomatology*. This indicator was measured through five items that ask about mood indicators in the past four weeks. For example, “How often have you felt little desire to do things?” (1 = never, 2 = rarely; 3 = sometimes; 4 = almost always; 5 = always). For this study, the scale showed adequate reliability (alpha = 0.85).

*Positive and negative affect.* This indicator was measured by using the PANAS instrument [46], which is widely used to assess positive and negative affect. It consists of 10 items of positive affect and 10 items of negative affect. Each item is marked from 1 (nothing) to 5 (very frequently). In Chile, it has shown adequate psychometric properties [47]. In this study, the scale presented alpha = 0.78 for positive affects and alpha = 0.73 for negative ones.

*Happiness.* This indicator was measured through the question: “In general, would you say you are...?” (1 = not happy at all; 2 = not very happy; 3 = pretty happy; 4 = very happy). This evaluation is commonly used with adult participants [48].

*Satisfaction with relationship*. This indicator was measured through the question: “How satisfied are you in your relationship?” (1 = not satisfied at all; 10 = completely satisfied).

*Satisfaction with sex life*. This indicator was measured through the question: “How satisfied are you with your sex life?” (1 = not satisfied at all; 10 = completely satisfied).

*Free time.* This indicator was measured through the question: “Do you have free time over the weekends?” (1 = always; 4 = never)

*Freedom to set life goals.* This indicator was measured through the question: “How much freedom would you say you have to set your own personal goals and achieve your life projects?” (1 = much freedom; 4 = no freedom).

### 2.3. Data Analysis

The R 3.6 software was used. In the main analysis, participants who had children were compared with those who did not in each one of the well-being indicators through an analysis of covariance. The sociodemographic variables (region, urban–rural, age, sex, socioeconomic status, educational level, having a partner) were considered as covariates to control. This analysis allowed us to obtain the estimated marginal averages for those who had children and those who did not, controlling for the sociodemographic variables, and then contrasts were made by these means. This analysis was repeated, including interactions with sex, age, and SES. Additional analyses were carried out when a significant interaction effect was detected.

The analyses were carried out with the whole sample and then repeated independently in the women’s and men’s samples.

The study’s complex sample structure was considered, and a weighting factor was applied by region, urban–rural area, sex, and age group to restore the population distribution of the data. Multiple imputations (10 bases imputed with 15 interactions) using fully conditional specification with the R mice package were used.

## 3. Results

Table 1, relative to the whole sample, shows the statistically significant differences toward greater well-being among those who have no children in all the indicators, with the exception of two (happiness and satisfaction with relationship do not show statistically significant differences between those with and those without children). In women, statistically significant differences are observed between mothers and non-mothers, which favor the latter in all indicators; on the contrary, between fathers and non-fathers, statistically significant differences are only observed in three indicators (suffering, positive affects, and freedom to set goals), which also favor non-fathers.

Table 2 shows the correlations observed between the different indicators. The highest correlation (0.62) was the one obtained between life satisfaction and the ladder of the best possible life.

Table 3 shows the comparison of measurements estimated for each well-being indicator in people with and without children, this time controlled by the sociodemographic variables. In the comparisons where there were no interaction effects, two statistically significant differences were observed: those who have children showed fewer positive affects and more negative ones than those with no children. In relation to three indicators (ladder of the best possible life, free time, and freedom to set goals), statistically significant effects were observed. For the ladder of the best possible life and free time, parenthood interacted with age and SES. For the freedom to set goals, parenthood interacted with age.

Figure 1 shows the interaction between parenthood and age in the ladder of the best possible life. A U-shape relation was observed that indicates that those who have children and are younger than 50 show lower values in this indicator than those older than 50. On the contrary, those with no children show higher values at younger ages, which decrease as age increases.

Figure 2 shows the interaction between parenthood and SES in the ladder of the best possible life. It can be observed that people with children have higher values in this indicator than those with no children in the highest SES (ABC1, C2, and C3). However, the opposite occurs in the lowest SES (D and E).

Figure 3 and Figure 4 show the interaction between age and parenthood and between SES and parenthood with regard to free time. Figure 3 shows that people with children have less free time than those with no children up until just over 30 years of age. Afterward, the differences decrease and disappear after 70 years of age. Figure 4 shows that, for SES, ABC1, C3, and D, people with children have less free time than those without children. However, the levels of C2 and E do not show statistically significant differences.

Figure 5 shows the interaction between parenthood and age with regard to the freedom to set goals. It can be observed that, in people with children, there is a linear relationship between age and the freedom to set goals; however, in people with no children, the relationship between age and freedom has a U shape, which is at its its highest around the age of 20, then decreases to the lowest between the ages of 50 and 60, after which it then increases again.

The previous analyses (with the same covariates and interactions) were repeated separately with the women’s and men’s samples (tables and figures not presented; if interested, request to the authors). In women, in comparisons where there were no interaction effects, it was observed that mothers presented statistically significant differences to non-mothers in suffering and negative affects (higher values), as well as positive affects and the freedom to set goals (lower values). Statistically significant interactions were also observed between maternity with age and SES in the ladder of the best possible life and between maternity with age in life satisfaction, between free time and age. These interactions were similar to those observed in the total sample. The interaction between maternity with age and SES in the ladder of the best possible life in the total sample indicated a lower satisfaction among mothers less than 50.

In the comparison of men with and men without children, it was observed that the former showed statistically significant differences with the latter in the ladder of the best possible life and negative affects (higher values), as well as positive affects (lower values). In the men’s sample, an interaction of paternity with age was observed in suffering, which slightly increased with age in those with children; however, in those with no children, an inverted U was observed, reaching its highest at the age of 50. There was also an interaction effect in free time, where it was observed that although this is always lower in men with children than in those without children, this difference increases with time.

## 4. Discussion

Being a parent can be a great source of satisfaction. It implies challenges that force the deployment of skills and learning and the acquisition of new roles and responsibilities that can expand one’s sense of personal identity and provide new meaning to many vital acts. However, parenthood can also pose significant difficulties and generate stressors, particularly in certain social conditions and life circumstances. The mode of social organization and prevailing cultural patterns can enhance some of these different facets of parenthood [49].

In the accelerated and demanding ways of life that tend to prevail in many contemporary societies, a question of particular relevance is whether parental responsibilities may be becoming particularly complex to perform. Parental stress and even parental burnout can significantly affect the well-being of parents, which has implications not only for them but also for the well-being and development of their children, for the birth rate, and for society as a whole [22].

The results obtained in this sample of national scope in Chile point in the same direction. Broadly speaking, it is observed that almost all well-being indicators are higher in those with children compared to those with no children. When controlling for the effect of the sociodemographic differences, those who are parents report experiencing fewer positive affects than those who are not and more negative affects, suggesting that the implications of parenthood in everyday life are emotionally exhausting.

Other well-being indicators interact with aspects such as SES and age. Those who have children and live in the least privileged SES show lower values in the ladder of the best possible life than those who have no children. On the contrary, positive effects of parenthood are observed in the ladder of the best possible life in the highest socioeconomic groups. These results suggest that social conditions and not having sufficient means to care for children are factors that negatively affect some aspects of the well-being of parents in Chile.

The relationship between free time and parenthood was also observed as differentiated by SES, though the results in this second case are less clear and harder to interpret.

The results obtained also show the relevance of taking into account age when considering the impact of parenthood on subjective well-being. At a younger age, the differences between those who have children and those who do not are higher in the ladder of the best possible life, as well as in free time and the freedom to set goals. The experience of being a young mother or father in our present social context appears to be particularly challenging. Therefore, it is interesting to analyze the full inversion of the relationship between subjective well-being (indicated by the ladder of the best possible life) and having children or not after 50 years of age, where those with no children appear to have clearly lower levels of well-being than those with children. Of course, there are numerous factors connected to this lower level of well-being, and it is not possible to establish any type of causality, let alone if we consider how highly selective the population who had no children was in the sociocultural context of past decades. The relationship between parenthood and well-being in the case of older people and in relation to children who are already adults is a very different phenomenon from what usually draws attention [15].

However, just as parenthood can imply greater stress, it can also imply great satisfaction and higher well-being, in a complex and dynamic balance affected by a wide variety of factors. Nomaguchi and Milkie [22] propose a demand-and-reward approach to analyze the relationship between parenthood and well-being and propose situating the determiners in the framework of the model of stress and from a vital cycle perspective. In a similar direction, Nelson, Kushlev, and Lyubomirsky [50] proposed a model of parental well-being where they highlighted four factors that favor the well-being of parents (increase in meaning in life, satisfaction of human psychological needs, positive emotions, broadening of social roles) and four others that affect them negatively (increase in negative emotions, financial strain, sleep disturbance, and strained partner relationships).

In many countries, the cultural trends have favored a greater instability of family ties, lower access to the traditional social support of the parental exercise such as a big family and a fast pace of life, which has made the exercise of parental roles more complex [51]. On the other hand, in recent decades, the expectations regarding these roles have increased. Some authors have posed the growing predominance of what they have called “ideology of intensive parenting” to refer to a conception of the maternal and paternal exercise of total commitment and dedication to the children, which would set disproportionate standards of what being a good parent would imply, increasing the feelings of guilt, failure, and self-demand [12,52].

A key factor in the analysis of the relationship between parenthood and well-being is gender. The overload that taking care of children usually implies for women is well established, and the negative effect it may have on various aspects of subjective well-being has been suggested by several studies [53]. Although changes have occurred toward the greater involvement of fathers in parenting, caring tasks continue to be strongly differentiated by gender, with an unequal burden being to the detriment of women [54,55]. The amount of time that mothers devote to the care of their children has not decreased; on the contrary, given their increasing incorporation into the working world, women with children have had to undertake additional tasks and roles [56]. Nonetheless, in this study, no effects of the interaction between parenthood and sex were observed in the indicators of well-being in the total sample; however, the global results comparing women with and without children to men with and without children showed more unfavorable indicators of well-being in mothers than in fathers. It is interesting that, unlike in other studies, where men with children appear to have higher levels of well-being than those with no children [57], in this study, being a father appears to be related to fewer indicators of well-being.

Two strengths of this study are the probabilistic and national nature of the database used and the systematic method of analysis used. Both general analyses, with no control of sociodemographic variables that could blur interpretations, and analyses with control of these variables were considered. In addition, special treatment was used to consider some potentially relevant interaction effects. Although the study does not allow us to draw causal conclusions regarding the relationship between parenthood and well-being, robust comparisons were allowed. The lack of a simultaneous consideration of these strategies of analysis has produced inconclusive controversies in the literature [14,18,58].

On the other hand, a limitation of this study is that there were no equivalent proportions of participants in the different sociodemographic groups. Another limitation is that most of the measures used were single-item. Although they have been shown to be useful [44], single-item measures tend to be less reliable, which attenuates associations with other variables. Another limitation of our study is that, although various sociodemographic variables (region, urban–rural, age, sex, socioeconomic status, educational level, and having a partner) were controlled, the analysis carried out did not allow us to focus on central aspects that have an impact on parenthood, such as the type of family structure or the type of relationship parents had with their children, among others.

An interesting aspect to consider is that, while the question about the ladder of the best possible life showed relationships with parenthood, the measure of life satisfaction, despite showing similar trends, did not show statistically significant relationships with parenthood. The ladder of the best possible life demands a greater cognitive effort and places the person in the framework of a more explicit analysis regarding their own expectations in life. Although both measures are closely related, it is not unusual for participants to respond in somewhat different ways [44]. Given that the correlation between the different indicators considered in this study was not very high and given the informative value of each one, we decided not to group them.

The existence of a gap in some dimensions of well-being between those who are parents and those who are not has been observed in studies in other countries. However, it is not a consistent result, with studies showing the opposite [59,60]. In Latin America in general and Chile in particular, there is little research in this regard, but results have also been observed to converge on the hypothesis of the well-being gap in parenthood [20]. While the existence of this gap, on some levels, is by no means incompatible with the existence of various significant and positive aspects in the experience of parenthood, the data provide a wake-up call regarding the prevailing way of life and social organization and about challenges that public policies should consider. Greater economic benefits for families, childcare support, and high flexibility in working hours have a clear positive impact on the well-being of those who are parents [18,40]. Along with the above, in our social context, policies aimed at reducing gender inequalities and the traditional distribution of gender roles seem particularly relevant.

This is of special interest in Chile, a Latin American country that has undergone a process of intense socioeconomic and cultural transformations in past decades. At an economic level, there has been significant development and a substantial reduction in extreme poverty that has been accompanied by the maintenance of high levels of inequality and a reduced role of the state in the provision of public goods [61]. In this scenario, a strong social conviction that achievements in life depend on personal effort has been created, favoring the withdrawal from the public into the private world (with the family world included in the latter) as a privileged and almost unique space of comfort and bonding [62]. Chilean society is characterized by significant tensions arising from this social context, with relatively high levels of life satisfaction coexisting at a subjective level alongside high levels of emotional and social unrest, and strong differences between all of these aspects depending on one’s socioeconomic status [29].

Chile shares (with other American nations) a relatively collectivistic culture, focused on familism, and with a relatively high distance from power [63] and the so-called “sympathetic” style of interaction, with an emphasis placed on avoiding conflict and maintaining positive interpersonal relationships [64]. These factors are potentially attenuators of the impact of the demands of parenting on the well-being of parents, as implied in a recent publication that links parental burnout to individualistic cultures [24].

## 5. Conclusions

According to various indicators of subjective well-being, those who have children in Chile appear to be in a less favorable condition than those who do not. In some of these indicators, it can be observed that an earlier age when having children and a lower socioeconomic status is related to an increase in the well-being gap between those who have children and those who do not. Likewise, more unfavorable indicators of subjective well-being were observed in mothers than in fathers. These results converge with other studies in different countries that suggest that having children can pose demands and create concerns that affect parents’ subjective well-being, particularly in certain conditions, suggesting the convenience of promoting modifications in the social traditions of understanding and supporting parenthood. Social policies that facilitate the parental role and its related demands, especially for those who are in less privileged socioeconomic conditions, and that promote the modification of traditional patterns in relation to the distribution of parental roles by gender, seem to be relevant strategies.

## Figures and Tables

**Figure 1 ijerph-18-07408-f001:**
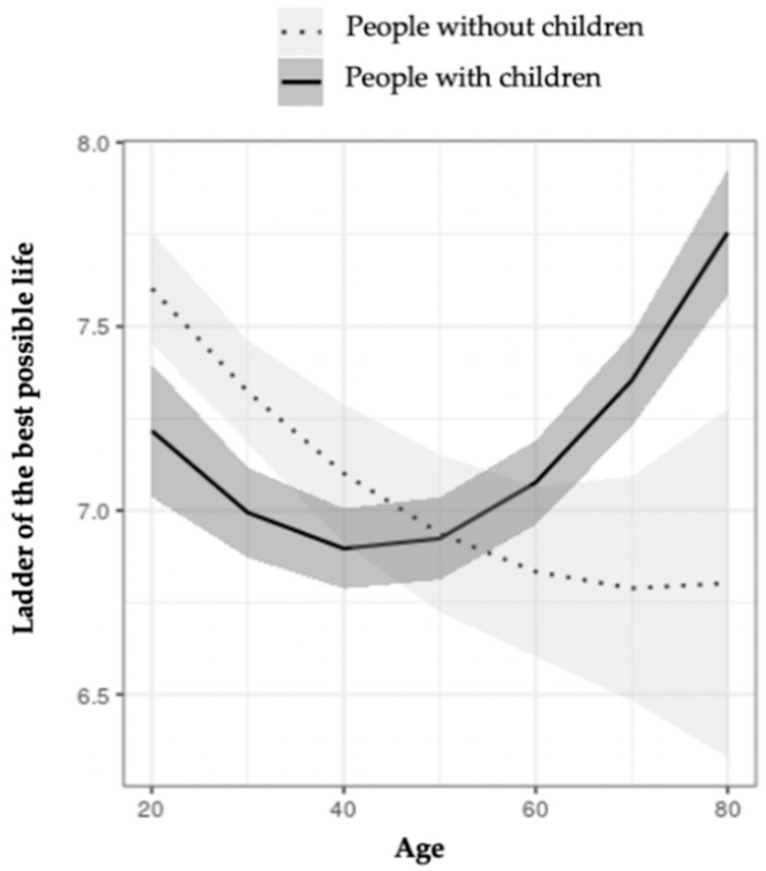
Moderation effect of age on the ladder of the best possible life in people with and without children.

**Figure 2 ijerph-18-07408-f002:**
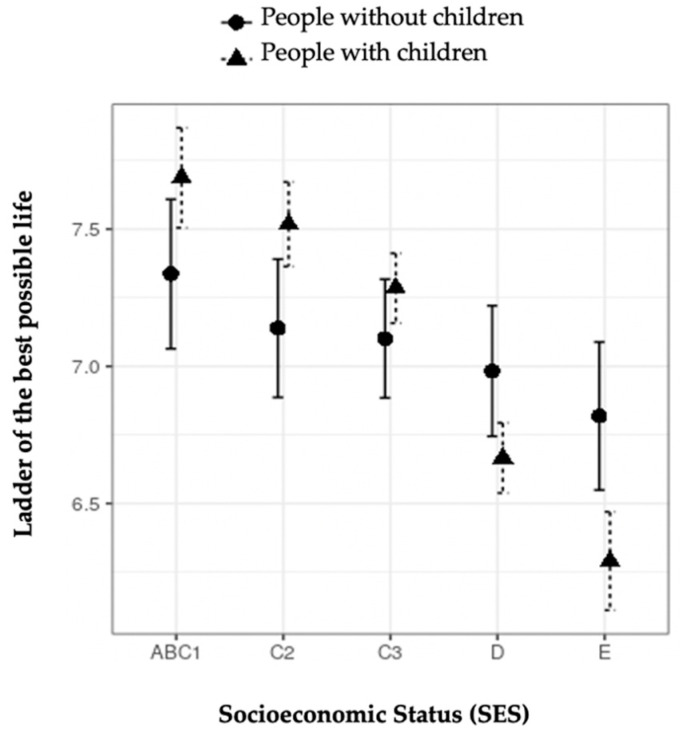
Moderation effect of socioeconomic status on the ladder of the best possible life in people with and without children.

**Figure 3 ijerph-18-07408-f003:**
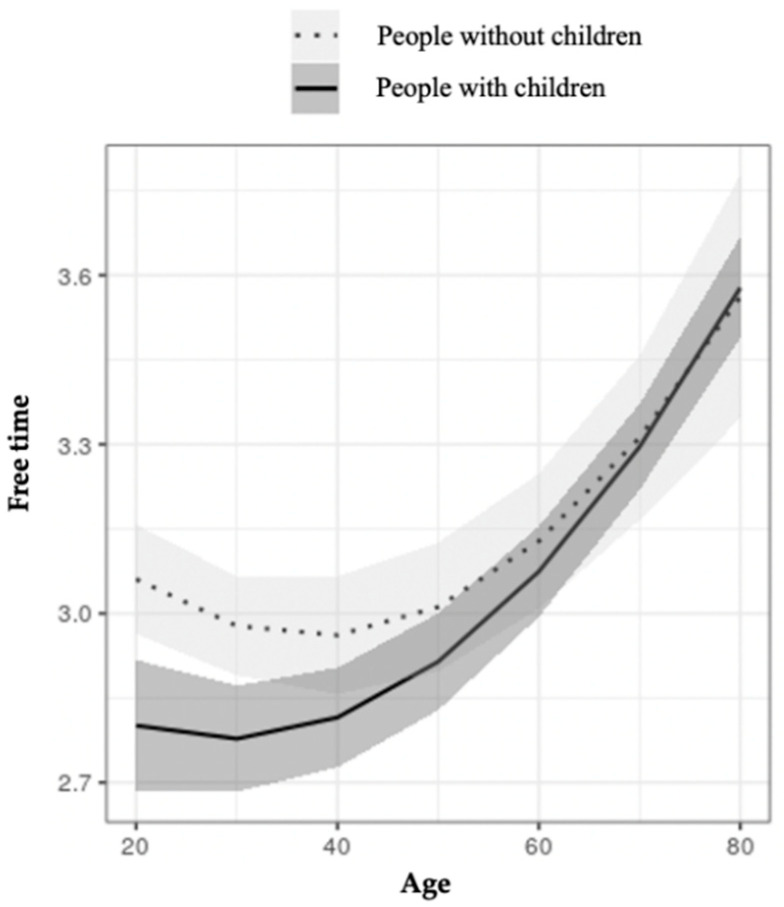
Moderation effect of age on free time in people with and without children.

**Figure 4 ijerph-18-07408-f004:**
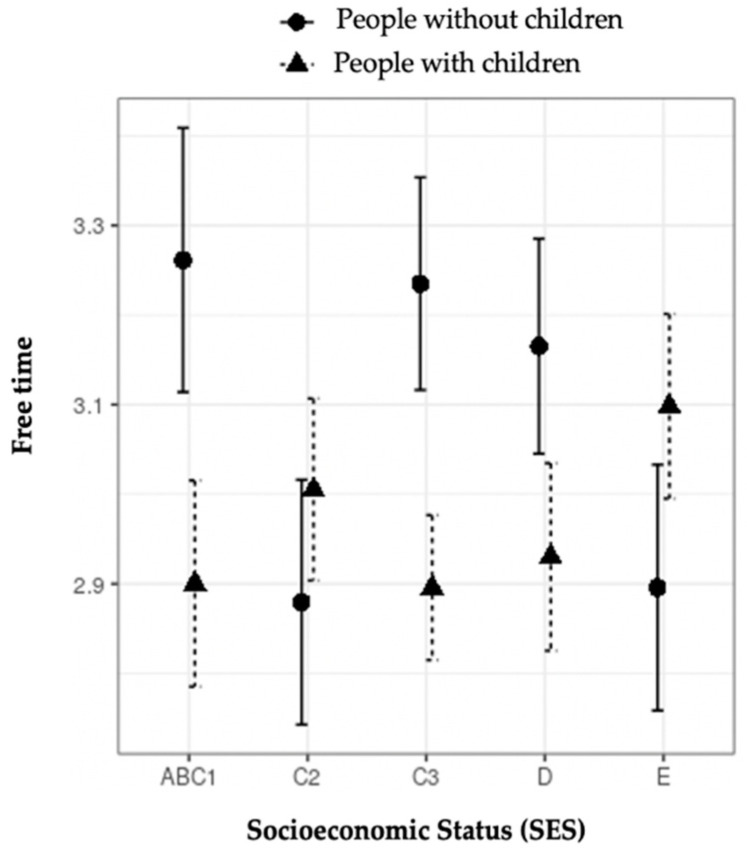
Moderation effect of socioeconomic status on free time in people with and without children.

**Figure 5 ijerph-18-07408-f005:**
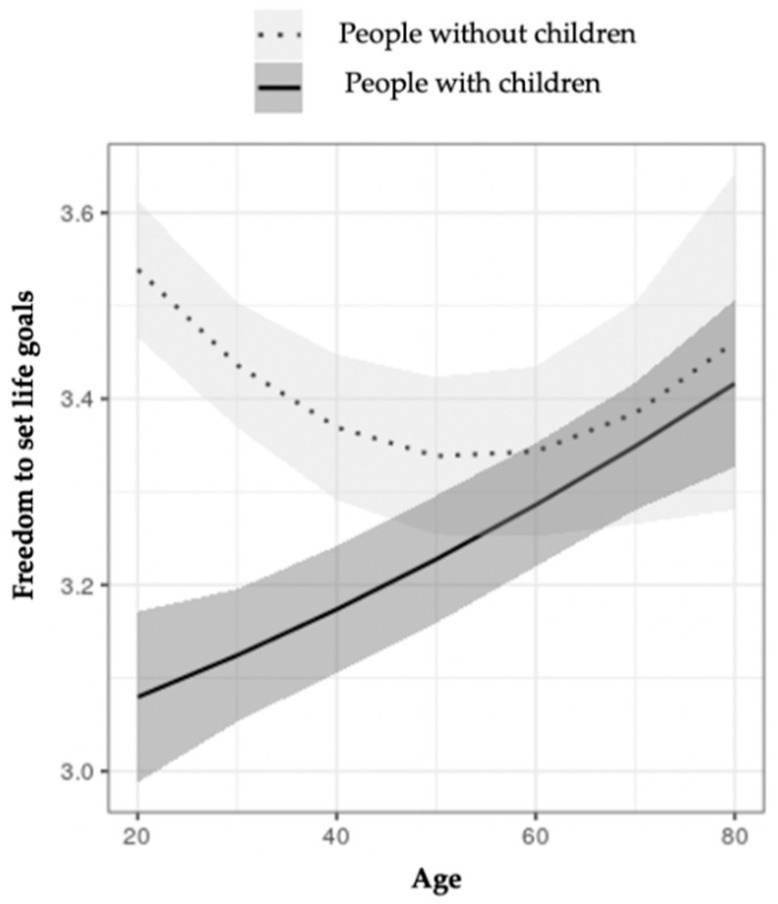
Moderation effect of age on the freedom to set life goals in people with and without children.

**Table 1 ijerph-18-07408-t001:** Differences by sex between participants with and without children.

	Men with and without Children							Women with and without Children							Men and Women with and without Children						
	With (Media)	SD	Without (Media)	SD	With and Without (Media)	SD	*p*	With (Media)	SD	Without (Media)	SD	With and Without (Media)	SD	*p*	With (Media)	SD	Without (Media)	SD	With and Without (Media)	SD	*p*
1. Life satisfaction	7.21	2.11	7.41	1.96	7.27	2.07		7.2	2.25	7.76	1.87	7.27	2.21	***	7.2	2.19	7.52	1.94	7.27	2.14	***
2. Ladder of the best possible life	6.9	1.97	7.1	1.99	6.96	1.97		6.87	2.11	7.48	1.89	6.95	2.09	***	6.88	2.05	7.22	1.97	6.95	2.04	***
3. Suffering	5.99	2.36	5.55	2.44	5.87	2.39	*	6.59	2.53	5.3	2.41	6.42	2.55	***	6.32	2.47	5.47	2.43	6.14	2.49	***
4. Depressive symptomatology	2.17	0.85	2.09	0.81	2.15	0.84		2.51	0.97	2.23	0.8	2.47	0.95	***	2.36	0.94	2.14	0.81	2.31	0.91	***
5. Positive affect	3.83	0.68	3.94	0.66	3.86	0.68	*	3.73	0.77	3.9	0.64	3.75	0.75	***	3.77	0.73	3.92	0.65	3.8	0.72	***
6. Negative affect	2.72	0.75	2.69	0.74	2.71	0.74		3.02	0.8	2.79	0.66	2.99	0.79	***	2.89	0.79	2.72	0.71	2.85	0.78	***
7. Happiness	3.1	0.68	3.06	0.67	3.09	0.68		2.97	0.76	3.14	0.66	2.99	0.75	**	3.03	0.73	3.09	0.67	3.04	0.71	.
8. SR ^a^	8.8	1.65	8.77	1.6	8.79	1.63		8.38	2.1	8.82	1.66	8.44	2.05	**	8.57	1.92	8.79	1.62	8.61	1.87	
9. Sex life	7.64	2.25	7.59	2.34	7.63	2.27		6.65	2.8	7.44	2.47	6.76	2.77	***	7.09	2.62	7.54	2.38	7.19	2.58	*
10. Free time	2.97	0.99	2.97	0.97	2.97	0.99		2.91	1.02	3.13	0.92	2.94	1.01	**	2.94	1.01	3.02	0.95	2.95	1	*
11. FSLG ^b^	3.23	0.87	3.42	0.74	3.29	0.84	**	3.07	0.93	3.48	0.71	3.12	0.91	***	3.14	0.91	3.44	0.73	3.2	0.88	***

Note: SD = standard deviation; ^a^ satisfaction with relationship; ^b^ freedom to set life goals; * *p* < 0.05, ** *p* < 0.01, *** *p* < 0.001.

**Table 2 ijerph-18-07408-t002:** Pearson’s correlation coefficients between the measured dimensions of well-being.

Well-Being Dimensions	1	2	3	4	5	6	7	8	9	10	12
1. Life satisfaction	1 **	0.62 **	−0.19 **	−0.36 **	0.35 **	−0.36 **	0.44 **	0.33 **	0.28 **	0.10 **	0.22 **
2. Ladder of the best possible life		1 **	−0.20 **	−0.34 **	0.35 **	−0.32 **	0.42 **	0.30 **	0.27 **	0.09 **	0.23 **
3. Suffering			1 **	0.27 **	−0.11 **	0.23 **	−0.20 **	−0.13 **	−0.13 **	0.00	−0.09 **
4. Depressive Symptomatology				1 **	−0.44 **	0.55 **	−0.39 **	−0.26 **	−0.26 **	−0.09 **	−0.26 **
5. Positive affect					1 **	−0.42 **	0.41 **	0.27 **	0.25 **	0.13 **	0.28 **
6. Negative affect						1 **	−0.35 **	−0.28 **	−0.19 **	−0.13 **	−0.22 **
7. Happiness							1 **	0.36 **	0.34 **	0.09 **	0.26 **
8. Satisfaction with relationship								1 **	0.38 **	0.09 **	0.18 **
9. Sex life									1.00 **	0.05	0.20 **
10. Free time										1.00 **	0.16 **
11. Freedom to set life goals											1.00 **

Note: ** *p* < 0.01.

**Table 3 ijerph-18-07408-t003:** Comparison between participants with and without children in different dimensions of well-being.

Well-Being Dimensions	Without Children	With Children	d	Test of the Difference between People with and without Children ^a^	Interaction between Having and Not Having Children with Socioeconomic Status and Age ^b^
1. Life satisfaction (LS)	7.7	7.47	0.11	F (1.000, 70634.595) = 2.518, *p* = 0.113	F (7.000, 880577.800) = 1.021, *p* = 0.414
2. Ladder of the best possible life	7.24	7.09	0.08	F (1.000, 25325.281) = 1.104, *p* = 0.293	F (7.000, 1101321.113) = 2.278, *p* = 0.026
3. Suffering	5.65	5.95	0.13	F (1.000, 27216.497) = 3.820, *p* = 0.051	F (7.000, 2120919.148) = 1.775, *p* = 0.087
4. Depressive symptomatology (DS)	2.12	2.2	0.09	F (1.000, 26554.745) = 1.871, *p* = 0.171	F (7.000, 1719087.595) = 0.857, *p* = 0.540
5. Positive affect (PA)	4.01	3.83	0.24	F (1.000, 41284.314) = 15.044, *p* = 0.000	F (7.000, 1825570.601) = 1.105, *p* = 0.357
6. Negative affect (NA)	2.56	2.74	0.24	F (1.000, 44297.595) = 13.278, *p* = 0.000	F (7.000, 3793673.359) = 1.126, *p* = 0.343
7. Happiness	3.07	3.09	0.02	F (1.000, 29844.828) = 0.118, *p* = 0.731	F (7.000, 830267.892) = 1.609, *p* = 0.127
8. Satisfaction with relationship	8.88	8.66	0.12	F (1.000, 15.322) = 1.472, *p* = 0.243	F (7.000, 446.394) = 0.655, *p* = 0.710
9. Sex life	7.3	7.08	0.09	F (1.000, 861.028) = 1.926, *p* = 0.166	F (7.000, 769.221) = 1.349, *p* = 0.224
10. Free time	3.14	2.97	0.18	F (1.000, 8577.039) = 7.254, *p* = 0.007	F (7.000, 910339.668) = 3.564, *p* = 0.001
11. Freedom to set life goals	3.45	3.23	0.26	F (1.000, 25616.367) = 15.429, *p* = 0.000	F (7.000, 754387.778) = 2.149, *p* = 0.035

^a^ Test of the difference between people with and without children, controlled by sociodemographic variables. ^b^ Test for the presence of interaction effects between having and not having children with socioeconomic status, sex, and age.

## Data Availability

Restrictions apply to the availability of these data. Data were obtained from PNUD and are available upon request to matias.cocina@undp.org, with the permission of Marcela Rios (Representative Resident Assistant of PNUD Chile).

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
