# Peer review of "Subjective Well-Being and Parenthood in Chile"

_ijerph, 2021, doi:10.3390/ijerph18147408_

Round 1

Reviewer 1 Report

Present manuscript aims to examine differences between parents and non-parents from Chile. Parents and adults without children. Parents and non-parents are compared in different adjustment criteria: Life Satisfaction, Ladder of best possible life, suffering, depressive symptomatology, positive affect, negative affect, happiness, satisfaction with couple life, sex life, free time, and freedom set life goals. Overall, it seems that adults without children have more adjustment than parents.

Present manuscript has important theoretical and empirical weaknesses that undermine the interest for potential readers. Although findings could be an interesting contribution to the literature, it should be correctly rationalized including within the text important previous classical and recent studies to adequately connect present manuscript with the literature. To test differences in adjustment and competence it is quite important to compare adults without children with different parents (for example, according to family structure: step-parent, single parent and both natural parents) rather than parents as a whole. It should be reviewed more detailed within the text important family variables according to previous classical and recent studies to improve the poor theoretical and empirical quality.

  • Introduction

An important variable widely identify within the literature has been neglected in the manuscript: parenting stress. Happily, for most parents the experience of parenthood is a positive one overall. However, it is documented that there are differences in parenting stress among parents, defined as negative feelings toward the self and toward the child or children, and by definition these negative feelings are directly attributable to the demands of parenthood. Parenting stress has a negative impact not only for children, but also for parents (see Deater-Deckard, 1998).

In the literature parents has been identified (i.e., indulgent, authoritative, authoritarian, and neglectful) according to their warmth and strictness (Queiroz, Garcia, Garcia, Zacares, & Camino, 2020). Differences among children and adolescents can be explained as a function of parenting (Yeung, 2021) For example, children with warm parents have more adjustment than their peers with non-warm parents (Fuentes, Alarcon, Garcia, & Gracia, 2015).

Additionally, the different family structure has an important consequence (benefit or harmful) for children and adolescents. It has been coherently related differences in competence and adjustment depending on family structures as both natural parents, single-parent, or stepparents (Steinberg, 1986, 1987).

  • Method
  •  

Authors should add the number of participants by age-range. Additionally, if authors have the data, it should be included the number of parents according to the type of family (e.g., single parent, stepparents, or both natural parents).

  • Measures

Authors should justify within the text the measures based on previous studies with the same or similar procedure. Happiness has measured with a single item, but this strategy is frequent and valid in studies with adults (e.g. Garcia, Fuentes, Gracia, Serra, & Garcia, 2020).

  • Discussion

Authors should discuss their results including a little bit more the limitations, specially that there are not considered any parenting variable (e.g., parenting style, type of family, or parenting stress) to compare with adults without children.

Importantly, especially parenting stress should be highlighted due to it clearly that parents, regardless their levels of parenting stress, tend to have more stress that adults without children, so present results could be agreeing to the literature (Deater-Deckard, 1998). Additionally, macrosocial variables could have an impact on parenthood, so it should be study in future studies (Gracia, García, & Musitu, 1995). Parenthood could be affected by sex roles and laws (Lila, Gracia, & García, 2013). In comparison with males, females suffer more sexism (Lila, Gracia, & García, 2010) and more physical violence (Gracia, García, & Lila, 2008). Therefore, it should be differentiated in future studies between mothers and fathers.

Additionally, the importance of cultural context should be highlighted a little bit more. Previous studies showed that the impact of parenting could not be always the same in all cultural contexts (Ridao, López-Verdugo, & Reina-Flores, 2021; Gallarin, Torres-Gomez, & Alonso-Arbiol, 2021). It is possible that the impact of parenthood in the competence and adjustment of parents could be different depending on the cultural context (Yeung, 2021).

Finally, an important strong point of present manuscript should be highlighted: the relatively large sample, which helps to achieve good statistical power (Pérez, Navarro, & Llobell, 1999).

References

Deater-Deckard, K. (1998). Parenting stress and child adjustment: Some old hypotheses and new questions. Clinical Psychology: Science and Practice, 5(3), 314–332. https://doi.org/10.1111/j.1468-2850.1998.tb00152.x

Fuentes, M. C., Alarcón, A., García, F., & Gracia, E. (2015). Consumo de alcohol, tabaco, cannabis y otras drogas en la adolescencia: efectos de la familia y el barrio. Anales de Psicología, 31, 1000-1007. doi:10.6018/analesps.31.3.183491

Gallarin, M., Torres-Gomez, B., & Alonso-Arbiol, I. (2021). Aggressiveness in adopted and non-adopted teens: The role of parenting, attachment security, and gender. International Journal of Environmental Research and Public Health, 18(2034), 1-16.   doi:10.3390/ijerph18042034

Garcia, O. F., Fuentes, M. C., Gracia, E., Serra, E., & Garcia, F. (2020). Parenting warmth and strictness across three generations: Parenting styles and psychosocial adjustment. International Journal of Environmental Research and Public Health, 17(7487), 1-18.   doi:10.3390/ijerph17207487

Gracia, E., García, F., & Lila, M. (2008). Police involvement in cases of intimate partner violence against women: The influence of perceived severity and personal responsibility. Violence against Women, 14, 697-714.   doi:10.1177/1077801208317288

Gracia, E., García, F., & Musitu, G. (1995). Macrosocial determinants of social integration: Social class and area effect. Journal of Community and Applied Social Psychology, 5, 105-119.   doi:10.1002/casp.2450050204

Lila, M., Gracia, E., & García, F. (2010). Police attitudes toward intervention in cases of partner violence against women: The influence of sexism and empathy. Revista de Psicología Social, 25, 313-323.   doi:10.1174/021347410792675570

Lila, M., Gracia, E., & García, F. (2013). Ambivalent sexism, empathy and law enforcement attitudes towards partner violence against women among male police officers. Psychology, Crime and Law, 19, 907-919.   doi:10.1080/1068316X.2012.719619

Pérez, J. F. G., Navarro, D. F., & Llobell, J. P. (1999). Statistical power of Solomon design. Psicothema, 11, 431-436.

Queiroz, P., Garcia, O. F., Garcia, F., Zacares, J. J., & Camino, C. (2020). Self and nature: Parental socialization, self-esteem, and environmental values in Spanish adolescents. International Journal of Environmental Research and Public Health, 17(3732), 1-13.   doi:10.3390/ijerph17103732

Ridao, P., López-Verdugo, I., & Reina-Flores, C. (2021). Parental beliefs about childhood and adolescence from a longitudinal perspective. International Journal of Environmental Research and Public Health, 18(1760), 1-17.   doi:10.3390/ijerph18041760

Steinberg, L. (1986). Latchkey children and susceptibility to peer pressure: An ecological analysis. Developmental Psychology, 22(4), 433–439. https://doi.org/10.1037/0012-1649.22.4.433

Steinberg, L. (1987). Single parents, stepparents, and the susceptibility of adolescents to antisocial peer pressure. Child Development, 58(1), 269–275. https://doi.org/10.2307/1130307

Yeung, J. K. (2021). Family processes, parenting practices, and psychosocial maturity of Chinese youths: A latent variable interaction and mediation analysis. International Journal of Environmental Research and Public Health, 18, 1-15.   doi:10.3390/ijerph18084357

Author Response

Estimado revisor,

Gracias por tus comentarios. Adjuntamos un documento de Word, en el formato sugerido, con las respuestas a cada punto que indicaste.

Consulte el archivo adjunto

Saludos cordiales,

Reviewer 2 Report

Thank you for providing a chance to review your manuscript. Overall, the paper is well-written, but major revisions are still needed. Overall, this is an interesting study and provides a better understanding of Subjective Well-Being.

1.abstract section.

The results 14 confirm that in several indicators the subjective well-being levels of those who are not parents are 15 better than who are not.

I'm confused by the meaning of your sentence.

  1. Methods section

    Have all the measures you used been validited for Chileans populations? Do you use a Chileans version or you translated them into Chileans? If not, evidence of its validity and reliability should be provided according to the data of the analyzed sample (e.g., CFA, McDonald's omega). In this sense, evidence must be offered that subjective well-being or state of mindfulness better fit a one-dimensional structure than a multidimensional structure. Of all the instruments, the Cronbach's alpha obtained in each subscale must be indicated, as well as some example item.

  It is better to introduce how you did data analysis more specifically. For example, how you merged the PANAS and HAPPINESS scores into SWB? and how you dealt with them in statistical approach more specificly? Did you use parcelling?

  1. Discussion section

Further discussion and interpretation of the effect of with children or with no children on SWB is necessary, which can be discussed from the perspective of cultural psychology. The culture of a country refers to the synthesis of common psychological meanings and social practices that distinguish a country from other countries. From the perspective of psychology, culture can shape its members' values, self-construal, think styles, social axioms, subjective well-being and many other aspects. These internalized cultural tendencies will have a series of influences on the individual's evaluation of subjective well-being. To take the simplest example, in countries that value individualism (as opposed to collectivism), members of society are more likely to focus on stressors closely related to one's career development, such as job insecurity, the difficulties of working from home, the emergence of new career development opportunities, and so on. In contrast, members in a collectivist culture also focus on social relations, such as the health and career development of family members, relatives and friends. The concept of family consanguinity and parent-child relationship between the East and the West is different, which needs to be further discussed.

  1. Reference

The reference section is too old, please cite more references in the last 3 years.

Author Response

Dear Reviewer, 

Thank you for your comments. We attach a Word document, in the suggested format, with the answers to each point you indicated.

Kind regards,
